# Info-Coevolution: An Efficient Framework for Data Model Coevolution

Ziheng Qin [1 2]  Hailun Xu [2 §]  Wei Chee Yew [2]  Qi Jia [1]  Yang Luo [1]  Kanchan Sarkar [2 ◇]  Danhui Guan [2 ◇]
Kai Wang [1 §]  Yang You [1 †]

## Abstract

Machine learning relies heavily on data, yet the continuous growth of real-world data poses challenges for efficient dataset construction and training. A fundamental yet unsolved question is: given our current model and data, does a new data (sample/batch) need annotation/learning? Conventional approaches retain all available data, leading to non-optimal data and training efficiency. Active learning aims to reduce data redundancy by selecting a subset of samples to annotate, while it increases pipeline complexity and introduces bias. In this work, we propose Info-Coevolution, a novel framework that efficiently enables models and data to coevolve through online selective annotation with no bias. Leveraging task-specific models (and open-source models), it selectively annotates and integrates online and web data to improve datasets efficiently. For real-world datasets like ImageNet-1K, Info-Coevolution reduces annotation and training costs by 32% without performance loss. It is able to automatically give the saving ratio without tuning the ratio. It can further reduce the annotation ratio to 50% with semi-supervised learning. We also explore retrieval-based dataset enhancement using unlabeled open-source data. Code is available at https://github.com/NUS-HPC-AI-Lab/Info-Coevolution/.

## 1. Introduction

Machine learning is inherently data-driven and has numerous practical applications. Currently, the dominant paradigms for large-scale dataset construction fall into two categories: (1) collecting and annotating data for fully supervised training, or (2) sourcing data from the web, followed by automated cleaning and weakly-supervised/unsupervised training, and subsequently fine-tuning models on downstream tasks with supervised data. However, both approaches incur significant costs for both annotation and training, making them viable only for a few well-resourced institutions. As a result, the construction of large-scale datasets remains an insufficiently democratized and decentralized process.

Before 2010, popular datasets such as MNIST (Deng, 2012) and CIFAR-10/100 (Krizhevsky et al., a;b) each contained 60,000 hand-annotated low-resolution images, reflecting the modest scale of early machine learning datasets. The lack of data limited the development speed of machine learning at that time. In 2010, ImageNet-1K (Deng et al., 2009) was publicly released with more than 1M human annotated samples. It was the first large-scale supervised dataset, leading to the development of many modern deep-learning algorithms and architectures, boosting the development of AI research today. An estimated cost of building ImageNet1-k is at least 0.4 million USD for labeling the total 1.28m images, and 4 million for 14m ImageNet-12k, excluding other costs. With cloud GPU today, training a deep learning neural network on ImageNet-1K with A100 GPU takes about $240 to $1400, while labeling an ImageNet-scale (million-scale) data still takes more than 100 times this cost, ranging from $24,000 to more than $1m depending on annotators' degree of proficiency. Constructing supervised data is usually more expensive compared to training with them.

Besides the supervised datasets, other types of datasets emerge later with corresponding training schemes. Bert (Devlin et al., 2019) and GPT (Brown et al., 2020) use large-scale unsupervised text corpus, such as BooksCorpus (Zhu, 2015) and Common Crawl (Raffel et al., 2020), to train with masked language modeling/next-token-prediction. CLIP (Radford et al., 2021) scale the training on weakly supervised (web-sourced image-text pair) data with cross-modal contrastive learning. Segment Anything (Kirillov et al., 2023) utilizes models and manpower together to increase the data scale and quality of segmentation data iteratively. These different types of data have various quality and cost.

◇ Project lead § Core contribution [1]National University of Singapore, School of Computing, 11 Research Link, Singapore [2]ByteDance Ltd, Singapore. Correspondence to: Ziheng Qin <zihengq@comp.nus.edu.sg>, Kanchan Sarkar <kanchan.sarkar@bytedance.com>, Danhui Guan <guandanhui@bytedance.com>, Yang You <youy@comp.nus.edu.sg>.

*Proceedings of the 42nd International Conference on Machine Learning*, Vancouver, Canada. PMLR 267, 2025. Copyright 2025 by the author(s).

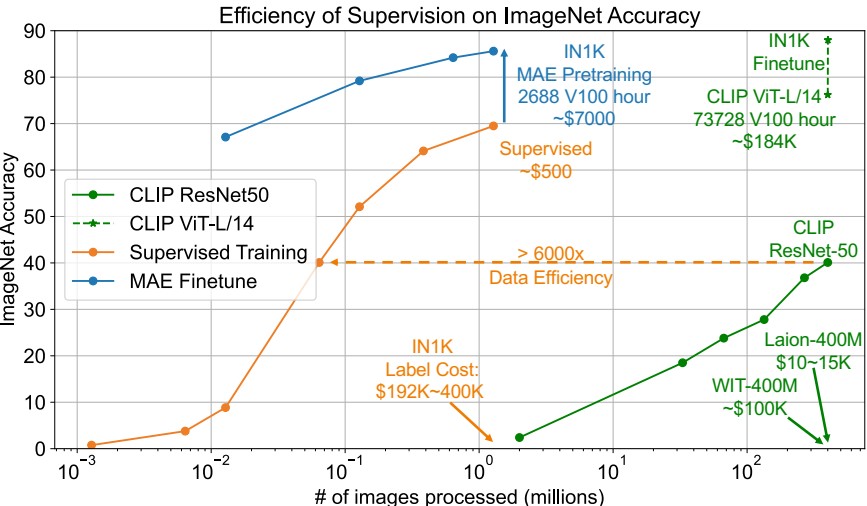

*Figure 1.* The data scaling curve of different types of data. Supervised data has higher data efficiency than weak supervised data on specific downstream tasks (e.g. ImageNet), while incurring a higher collection and annotation cost.

As shown in Fig.1, regarding the task-specific/downstream data efficiency(performance against data amount) and data construction cost, supervised data > weakly supervised data > unsupervised/self-supervised data. The data scale and training cost are in a reversed order, where supervised << weakly supervised < unsupervised/self-supervised. The three paradigms provide different tradeoffs between data collection/annotation and training costs. In general, a higher degree of supervision provides better data efficiency for downstream tasks with higher data cost per unit. Previously works like BLIP(Li et al., 2022) and Segment Anything (Kirillov et al., 2023) tried to boost the data quality with human+model annotation, which is somehow equivalent to improving the degree of supervision.

In online and downstream real-world scenarios, data continuously grows over time. The scale of unsupervised data could be immense, but the marginal gains from training diminish rapidly. Boosting its scale is not cost-efficient. Weakly supervised data involves a smaller overall volume with higher training efficiency per unit of data. In contrast, supervised data generally has the smallest total volume but incurs the highest annotation costs, leading to the most significant performance improvements for the same amount of data. As data grows in online scenarios, and usually the annotation cost is higher than training, it would be beneficial to improve the annotation efficiency and exlore the marginal benefits of supervised data.

To boost the annotation/sample efficiency, active learning was proposed, which aims to select samples that better benefit training to annotate. However, many of the current active learning methods(Li et al., 2024) involve frequent model

training with rounds of annotation, which makes the pipeline too complex for real-world applications. What's more, many active learning methods rely solely on the model's prediction for the sample's usefulness. It is prone to sample distribution problems on harder tasks. Their design of re-training + re-inference limits its application to large-scale data. Due to these limitations, active learning is not yet an efficient and robust enough solution for real-world scenarios. Coreset selection methods (Guo et al., 2022), on the other hand, skip the model update and leverage the sample distribution to select samples used for training. Therefore, it fails to leverage the model-specific information unless a model trained on fully annotated data is obtained initially.

To address these challenges, we propose Info-Coevolution, a novel and efficient online framework for selective data collection/annotation that integrates model-specific estimation with distribution awareness. Info-Coevolution enhances data annotation efficiency with minimal computational overhead. Leveraging Bayesian principles and our analysis of data locality, we propose a data-based information gain estimation strategy and Bayesian Prediction Fusion. This approach improves the model-specific sample selection while reducing the need for frequent model updates during the selection process. The framework begins with an unsupervised pretrained backbone and a small, randomly initialized dataset, mimicking real-world scenarios. By utilizing an online Approximate Nearest Neighbor (ANN) structure, such as HNSW (Malkov & Yashunin, 2018), Info-Coevolution achieves logarithmic scaling for information gain estimation, enabling it to efficiently handle growing data and increasing selective annotation.

On ImageNet-1K, Info-Coevolution achieves lossless performance with only 68% of the annotation cost, through just 4 rounds of continual supervised training (sum to <100% full training cost; training from scratch gets the same lossless result). The computational overhead primarily arises from model inference after each training round, totaling approximately 10 A100 GPU hours for the entire process. Additionally, we developed an efficient batch sampling mechanism, allowing the selection process to be completed within 1 minute on datasets at a million-scale. Furthermore, Info-Coevolution provides both qualitative and quantitative estimates of a data sample's information gain, conditioned on the model and the previously collected data. This feature enables automatic stopping when performance gains plateau, eliminating the need for additional sample annotations to verify saturation.

## 2. Related Works

**Coreset Selection Methods** Coreset selection methods focus on filtering out low-quality or redundant samples, while reserving the most representative ones in the target dataset. The core of these approaches lies in the elaborate selection criteria, including geometry-based (Sener & Savarese, 2017; Sinha et al., 2020) , error-based (Toneva et al., 2018; Paul et al., 2021) , decision-boundary-based (Ducoffe & Precioso, 2018; Margatina et al., 2021) , uncertainty-based (Coleman et al., 2019) , gradient-matching (Mirzasoleiman et al., 2020; Killamsetty et al., 2021a), bilevel optimization (Killamsetty et al., 2021b) and submodularity-based methods (Iyer et al., 2021; Zhou et al., 2023). Some of the approaches (Sener & Savarese, 2018) also combined the strength of active learning to make further improvements. In general, coreset selection methods do not force a strict constraint on full data/label access.

**Active Learning and Semi-Supervised Learning** Active learning (Hino, 2020; Smith et al., 2023; Li et al., 2024) and semi-supervised learning (Sohn et al., 2020; Zhang et al., 2022; Wang et al., 2023; Cai et al., 2022) are two complementary strategies for reducing the reliance on large-scale supervised datasets in machine learning. Active learning focuses on iteratively selecting the most informative samples for annotation, enabling models to achieve higher performance with fewer labeled examples by optimizing the data-labeling process. On the other hand, semi-supervised learning leverages a large pool of unlabeled data alongside a smaller labeled subset, using techniques such as pseudo-labeling, consistency regularization, or generative models to propagate label information and improve generalization. Together, these paradigms aim to maximize model efficiency and performance in data-scarce scenarios, often bridging the gap between fully supervised learning and real-world constraints. Segmenta anything, in a way similar to active

learning, conducts full annotation each round with different quality and re-train the model for each round on an increased amount/portion of data. Different to these works, our method targets an efficient online selective data collection/annotation process, with far less cost in the loop. An illustration of the difference is shown in Fig.2.

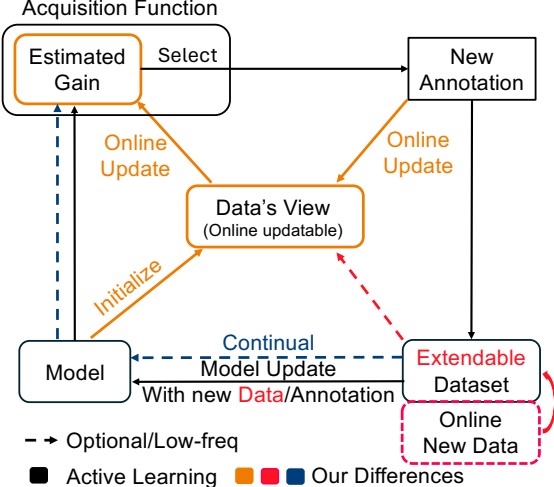

*Figure 2.* A comparison of our method's difference to active learning. Generally, active learning loops over 1. selecting samples, 2. annotating, 3. updating the model, and selecting samples again (with the updated model). In contrast, our method doesn't have to update the model frequently. We can update the model optionally with continual supervised learning at a much lower frequency (so the training cost is also low). And our algorithm is automatically class-balanced due to distribution awareness, therefore it doesn't suffer distribution shift as active learning.

## 3. From the Perspective of Information Gain

In this section, we first derive from the theoretical aspect of how to estimate the information gain of a sample to a dataset. Previously, an exact information gain was mainly defined upon an attribute of a decision tree. Here we extend it to a more generalized form and provide a way to estimate it on broader tasks, e.g. a sample's information gain with respect to image classification (datasets). We analyze the influence of locality in the space to improve both the estimation quality and efficiency. Then we formulate the selective data selection/annotation problem as a recursive online problem, each time adding one (batch of) sample/annotation to an already collected dataset. Based on this problem setup and our theoretical analysis, we propose our algorithm Info-Coevolution for efficient online selective annotation and dataset construction.

**Theorem 3.1.** *For neural networks that generate representation and then make the prediction (for model $M = g \circ f$), at certain distances $\epsilon$ from a sample point $x$ in*

*the feature representation space, the similarity of samples in feature space also leads to similar model predictions* $(\forall \delta, \exists \epsilon \ s.t. \forall x, x', |f(x') - f(x)| \leq \epsilon \Rightarrow |g(f(x')) - g(f(x))| \leq \delta)$.

We show a proof of Theorem 3.1 with both L2 distance and cosine distance in Appendix. This theorem implies a kind of linearity among near-neighbor samples' predictions, and explains one of the reasons that KNN predictions work. We will leverage this theorem to design a parametrized estimation of a sample's information gain with locality, extendability, and efficiency.

### 3.1. Information carried by sample to the dataset

In predictions with a decision tree $T$, information gain $IG$ with condition $A$ is defined as:

$$IG(T, A) = H(T) - H(T|A), \qquad (1)$$

which is the reduced entropy given condition A. Here, we extend this information gain to broader tasks. Assuming the samples we are curious about (target distribution) is from distribution $\rho$, which is either uniform in the whole distribution space or IID to train data, then the information gain of adding a sample $z = \{z_x, z_y\}$ to a dataset $\mathcal{D}$ sampled from $\rho$ is:

$$
\begin{aligned}
IG(\mathcal{D}, z) &= H(\mathcal{D}) - H(\mathcal{D} + z) \\
&= \int_{x, \|x-z\| \leq \epsilon} \big(H(x) - H(x|z)\big)\rho(x)\,dx,
\end{aligned}
$$
(2)

$$
\begin{aligned}
\text{or } IG(\mathcal{D}, z) &= H(\mathcal{D}) - H(\mathcal{D} + z) \\
(if\ IID) \quad &= \mathbb{E}\left[\frac{1}{|\mathcal{D}|} \sum_{x, \text{sim}(x,z) \geq \delta} \big(H(x) - H(x|z)\big)\right].
\end{aligned}
$$
(3)

Where $\epsilon \in \mathbb{R}$ is the distance threshold we decide to use, and $\delta \in [0, 1]$ is a similarity threshold. This distance threshold is inherited from Theorem 3.1, where two sample's prediction has a low correlation beyond a certain distance. During estimation, the $H$ here is estimated on parameterized $M_\theta$, and the $|| \cdot ||$ is defined within space of $f$.

Then a question is how to estimate $H(x|z)$ given $x \neq z$. As assumed, in certain distances the linearity dominates so that we can do interpolation and linear combination on logits/predictions. Within this space where linearity holds, interpolation between $z$ and $x$ where $y_z = y_x$ also gives the same label prediction, but with an increased certainty. Interpolation between $z$ and $x$ where $z_y \neq x_y$ will split the prediction over the two, which needs careful calculation. However, the interpolation itself is an estimation based on linearity assumption. As different samples' predictions

could have different certainty/confidence, we should take the confidence (probability of prediction being true) into consideration of the interpolation.

### 3.2. Confidence of label

By defining a confidence $\alpha$ for the prediction of a sample, we can integrate this value into the interpolation process as a weighted NN prediction:

$$\mathbf{y_{estimation}}(\mathbf{z}) = \frac{\sum_{x, sim(x,z) \geq \delta} \alpha_x Sim(x, z)\mathbf{y_x}}{\sum_{x, sim(x,z) \geq \delta} \alpha_x Sim(x, z)} \quad (4)$$

where $\mathbf{y_x}$ is the probability prediction vector of x's annotation. Then the entropy of a point z from data's view can be calculated based on it. Moreover, the semi-supervised learning's label and unlabeled datasets can be transferred into a continuous space defined on confidence $\alpha \in [0, 1]$, and we can estimate the confidence of an annotator and use models to perform annotation. An easy mapping in probability space can directly map accuracy to confidence: $\alpha = (acc - acc_{random})/(1 - acc_{random})$. One primary goal of semi-supervised learning under this interpretation could be making the $\alpha$ value converge while training the model with corresponding confidence without collapse.

### 3.3. Bayesian prediction fusion

Therefore, one key question in semi-supervised learning is interpreted as $\alpha$ value convergence. Suppose we have two independent predictors with confidence $\alpha_1$ and $\alpha_2$, when the two predictors have same prediction $y$ for sample $x$, the confidence value $\alpha_y$ of sample $x$ should be annotated with $y$ become

$$
\begin{aligned}
\alpha_y &= P\left(\frac{Predictions\ are\ true}{Predictions\ match}\right) \\
&= \frac{\alpha_1 * \alpha_2}{\alpha_1 * \alpha_2 + P(match\ with\ other\ label)}
\end{aligned}
$$
(5)

In classification tasks, $P(match\ with\ other\ label) \in [0, (1-\alpha_1)(1-\alpha_2)]$ depending on distribution over classes. In the binary classification case $P(match\ with\ other) = (1-\alpha_1)(1-\alpha_2)$, it requires smaller prediction confidence is larger than 0.5 so that the sample confidence is increased: $min(\alpha_1, \alpha_2) > 0.5 \Rightarrow \alpha_{xy} > max(\alpha_1, \alpha_2)$; if assuming the matching distribution is uniform over $c$ classes, then $P(match\ with\ other) = (1-\alpha_1)(1-\alpha_2)/(c-1)$ and $min(\alpha_1, \alpha_2) > 1/c$ is the condition. If no matching prob distribution is in other classes, then its value is 0. To sum up, using $(1-\alpha_1)(1-\alpha_2)$ gives a lower bound estimation on $\alpha_{xy}$. These conditions can be utilized to make the alpha value converge, or re-estimate $\alpha$ value for samples in a dataset.

When predictions from different views diverge, if one is confident above the threshold and the other is not, use the max

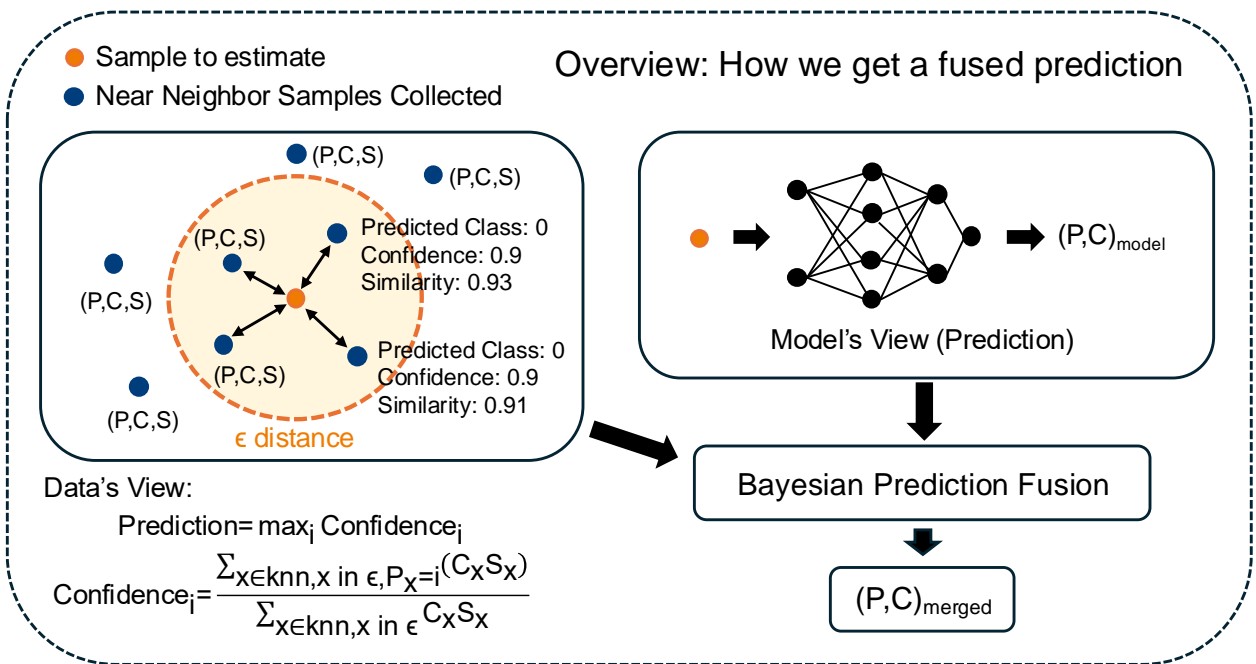

*Figure 3.* The overview of how we estimate a sample's confidence with Bayesian Fused prediction from both model's view and data's view. Generally, this estimation gives a better evaluation of a sample's uncertainty/confidence than using model prediction only.

one for the highest confidence. If both predictions are confident, then the sample's confidence is reduced. Supposing $\alpha_1 > \alpha_2$, then

$$
\begin{aligned}
P(p_1) &= \frac{P(p_1 \ is \ true)}{P(predictions \ unmatch)} \\
&= \frac{\alpha_1(1-\alpha_2)}{\alpha_1(1-\alpha_2) + P(other)}
\end{aligned}
\tag{6}
$$

where $P(other) \in [0, (1-\alpha_1)\alpha_2]$. In the case $p_2$ takes all the remaining confidences from the first predictor, $P(other) = (1-\alpha_1)\alpha_2$; if assuming the matching distribution is uniform over c classes, $P(other) = (1-\alpha_1)\alpha_2/(c-1)$. If there are no prob distribution unmatching classes on other classes, $P(other) = 0$. Similarly, using $P(other) = (1-\alpha_1)\alpha_2$ gives a lower bound on the updated confidence. The prediction divergence from confident predictions means $max(p_1) \neq max(p_2) \Rightarrow H(p') > max(H(p_1), H(p_2))$. Reducing the corresponding confidence can help with re-convergence.

### 3.4. Benefit of reannotating one sample

As the model is to absorb the data distribution, the dataset itself also carries a prediction based on the near neighbor prediction within the distance where linearity holds. And sample's confidence changes during turns of iteration. Therefore, taking the derivation from previous sections, when

re-annotating a sample, the expected gain is

$$
\begin{aligned}
\mathbb{E}[IG(annotate \ z)] = max\big(min(H(\alpha_z), \\
H(knn \ predictions)) - H(\alpha_{ann}), 0\big)
\end{aligned}
\tag{7}
$$

If using entropy, one can use $H(\alpha) = -\alpha log(\alpha) - \sum_{c-1} \frac{1-\alpha}{c-1} log\frac{1-\alpha}{c-1} \simeq -\alpha log(\alpha) - (1-\alpha)log(1-\alpha)$ as an estimation neglecting the constant term. In practice, we found using $H(\alpha) = \alpha$ is a feasible proxy to choose samples with less computation and higher compatibility. For sample selection, we only need to know the positivity and relative magnitude of the samples' information gain, therefore the confidence itself is a good proxy.

### 3.5. The Algorithm

**Selective Annotation** When data to be annotated is greater than the capacity, for a sample $z = (x_i)$, to estimate the gain of annotating it given already collected data D, we have model $m$ and get the feature $f(z)$, model prediction $y_m$, confidence $c_m = (p_m - 1/num\_class)/(1 - 1/num\_class)$. We then retrieve k nearest neighbors of z (with f and ANN search) and use Eq.4 to get a (knn) data-based prediction $y_{knn}, c_{knn}$. Then we merge two predictions from model and dataset with Eq.5 and Eq.6 and get $y_{merged}, c_{merged}$. The annotation gain of sample z is then calculated as $c_{ann} - c_{merged}$ where $c_{ann}$ is the average accuracy of the annotator. For batched selection, one can sample with probability proportional to the gain and drop redundant samples.

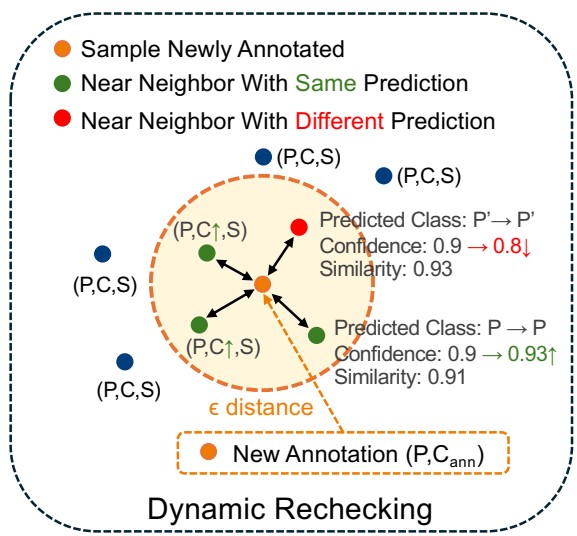

*Figure 4.* The idea of dynamic rechecking. When we get new annotation(s), we can update the sample estimation within the $\epsilon$-distance to better reflect the gain. It will automatically balance classes and sample density. This step is efficient with $k\log(n)$ time.

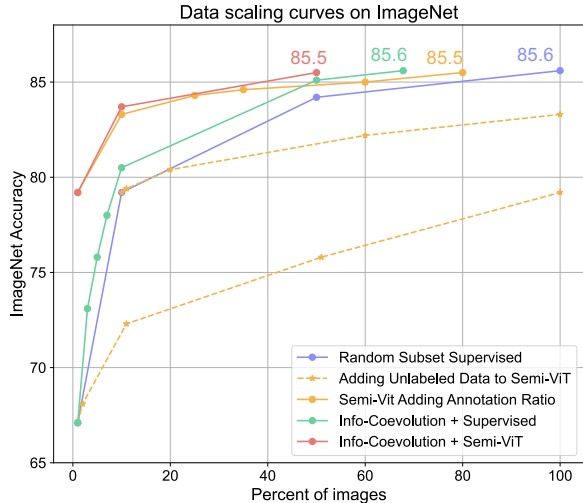

*Figure 5.* The scaling curve of different schemes. Info-Coevolution improves data efficiency for both Supervised/Semi-supervised settings, and gets lossless performance at 68% and 50% annotation ratios respectively.

**Dynamic Rechecking** After getting a new annotation $y_i$ for $z = (x_i)$ with confidence $c_{ann}$, we retrieve k nearest neighbors of z. For neighbors within the distance threshold, we recalculate their data's view prediction and update their (P,C) with our Bayesian Prediction Fusion Eq.5 and Eq.6 and update their gain. Each time we select the (batch) sample with the highest annotation gain to annotate. This design is to avoid redundancy and ensure class balance during selective annotation, as near samples with similar predictions will be updated with a lower gain, while near samples with different predictions will be updated with a higher gain. We use online-ANN based near neighbour search so that this process is efficient and extendable.

### 3.6. Using Public Data to Enhance Downstream Task

To further benefit downstream tasks and evaluate our method in the data/annotation growth setting, we construct a superset from multiple open-sourced (and mostly web-sourced) datasets (CC3M(Sharma et al., 2018), CC12M(Changpinyo et al., 2021), SBU(Ordonez et al., 2011), COCO(Chen et al., 2015), Visual Genome(Krishna et al., 2016), Laion-400m(Schuhmann et al., 2021)) and retrieve related samples to enhance downstream tasks. To reduce the peak GPU/RAM memory, we subsample 1% samples from all the samples and retrieve their feature using BLIP(Li et al., 2022). Then we use K-means to get 500 clusters. We then follow (Qin et al., 2024) to do ANN-based de-redundancy and filtering within each cluster and construct the corresponding ANN index for later retrieval. This allows a higher parallelism of the algorithms. For future data scale ex-

tension, either distributed HNSW or hierarchical recursive clustering plus cluster-wise HNSW is a feasible solution.

On ImageNet-1K, we use image embedding from ImageNet-1K samples to retrieve the k nearest neighbor in the superset, and then de-redundant the samples while filtering retrieved samples with cosine distance larger than 0.2.

## 4. Experiments

### 4.1. Datasets and Implementation Details

We evaluate our method on ImageNet-1K (Deng et al., 2009), CIFAR10/100(Krizhevsky et al., a;b), StanfordCars, Food-101(Bossard et al., 2014), SVHN(Netzer et al., 2011) under different annotation ratios and settings. We further extend the training data with data from CC3M, CC12M, SBU, Visual Genome, COCO, and LAION-400m to study the effect of scaling unlabeled data.

**Implementation Details.** The experiments on ImageNet-1K follows Semi-ViT (Cai et al., 2022), using ViT (Dosovitskiy et al., 2021) model with MAE (He et al., 2021) pretrained backbone to conduct supervised and semi-supervised training. All other details can be found in the Appendix.

### 4.2. Data/Annotation Scaling

**ImageNet-1k Results** We show our improvement in data/annotation efficiency here in Fig.5 and Tab.1. On 10% data setting of ImageNet, our selective annotation can increase the accuracy by 1.3% compared to the random baseline. With **only 68%** annotated samples from ImageNet-1k, we can achieve **lossless** performance (85.6% Acc), surpass-

ing the 85.5% Acc of Semi-ViT with 80% labeled data and 20% unlabeled data. What's more, we have an **automatic stop criterion** when the marginal gain of annotating more samples is negligible. The 68% ratio is given by the algorithm itself when it suggests to stop selecting samples, instead of a annotation ratio tuning. It can be seen that Info-Coevolution is compatible with Semi-Supervised learning, where Semi-ViT trained with 50% ImageNet-1K annotations selected by Info-Coevolution can achieve an almost lossless result (85.5%). Moreover, Info-Coevolution is compatible with continual supervised training, without introducing a distribution shift.

*Table 1.* Accuracy of training ViT-Large on ImageNet-1k with continually increasing annotation and doing continual training. Info-Coevolution gets a 1.3% accuracy improvement in 10% ImageNet supervised setting with the proposed prioritized selective annotation.

| Method | Setting | 1% | 3% | 5% | 7% | 10% |
|--------|---------|------|------|------|------|------|
| Random | Supervised | 67.1 | 72.5 | 74.6 | 76.6 | 79.2 |
| Ours | Supervised | - | 73.1 | 75.8 | 78.0 | 80.5 |

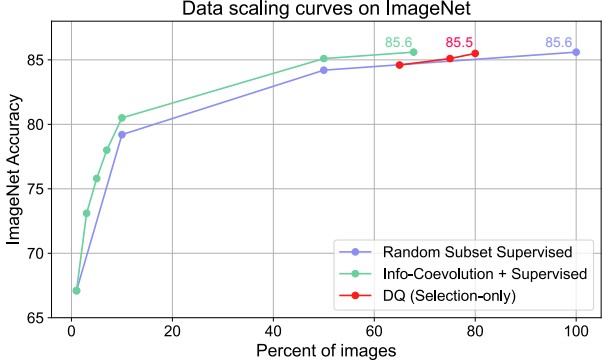

*Figure 6.* Compare with corset-selection SOTA method Dataset Quantization on IN1K.

**Comparison with Coreset Selection** In Fig.6, we compare with previous SOTA method Dataset Quantization(Zhou et al., 2023). With selection-only (not adding the MAE reconstruction), DQ achieves 85.5% Acc with 80% labeled data. It shows that beyond the extendability of Info-Coevolution, it can also serves as a good coreset selection method.

Tab.2 illustrates the effect of data scaling for further scaling semi-supervised training with unlabeled data. With additional unlabeled data, the acc can be further increased by 0.4% with semi-supervised training. For unlabeled data selection, we adjust the equation to better capture useful unlabeled data. See Appendix for detail. It use half of the unlabeled data to enhance the performance as using all unlabeled data.

*Table 2.* Scaling the dataset with retrieved data from our constructed superset and training with Semi-ViT. Extending the unlabeled data can improve performance while using our selection would improve the data efficiency of unlabeled data.

| Labeled Data | Additional Data | Acc |
|--------------|-----------------|------|
| 1.28M | 0 | 85.6 |
| 1.28M | Unlabeled 2M | 86.0 |
| 1.28M | Our selected unlabeled **1M** | 86.0 |

**Generalization and Robustness** Theoretically, the effectiveness of Info-Coevolution is model-agnostic and dataset-agnostic. We here verify its generalization and robustness across different datasets, architectures and other semi-supervised framework.

We present our experimental result on CIFAR10 semi-supervised training with Fixmatch(Sohn et al., 2020) and ResNet-50x4 in table 3. Info-Coevolution improves the data efficiency and further reduces the annotation amount to achieve lossless performance (95.85% acc) from 4000 to 1000.

*Table 3.* ResNet-50x4 with Fixmatch on CIFAR10

| Data Selection | 250 | 1000 | 4000 | Full |
|----------------|-------|-------|-------|-------|
| Random | 94.95 | 95.59 | 95.85 | 95.85 |
| Info-Evolution | 95.39 | **95.85** | - | - |

To analyze the generalization of Info-Coevolution across different datasets, we use Info-Coevolution with both supervised training and semi-supervised training by adapting pretrained ViT-L on CIFAR-10, CIFAR-100, Standfordcars, fool101 and SVHN. As shown in Fig.7, Info-Coevolution consistently improved both supervised and semi-supervised training performance on all these datasets.

### 4.3. Ablation Experiments

*Table 4.* Ablation of proposed components in the framework (1% model to select 10% annotation).

| Component | | | ImageNet Acc. |
|-----------|------|---------|---------------|
| Model | Data | Dynamic | 10% label |
| ✓ | | | 78.5$_{\downarrow 0.7}$ |
| | ✓ | | 79.8$_{\uparrow 0.6}$ |
| ✓ | ✓ | | 77.1$_{\downarrow 2.1}$ |
| ✓ | | ✓ | 79.7$_{\uparrow 0.5}$ |
| | ✓ | ✓ | 79.8$_{\uparrow 0.6}$ |
| ✓ | ✓ | ✓ | **80.2**$_{\uparrow 1.0}$ |
| Random Select | | | 79.2 |

**Ablation of Components.** As our algorithm involves fusing the prediction of model and KNN predictions with dynamic

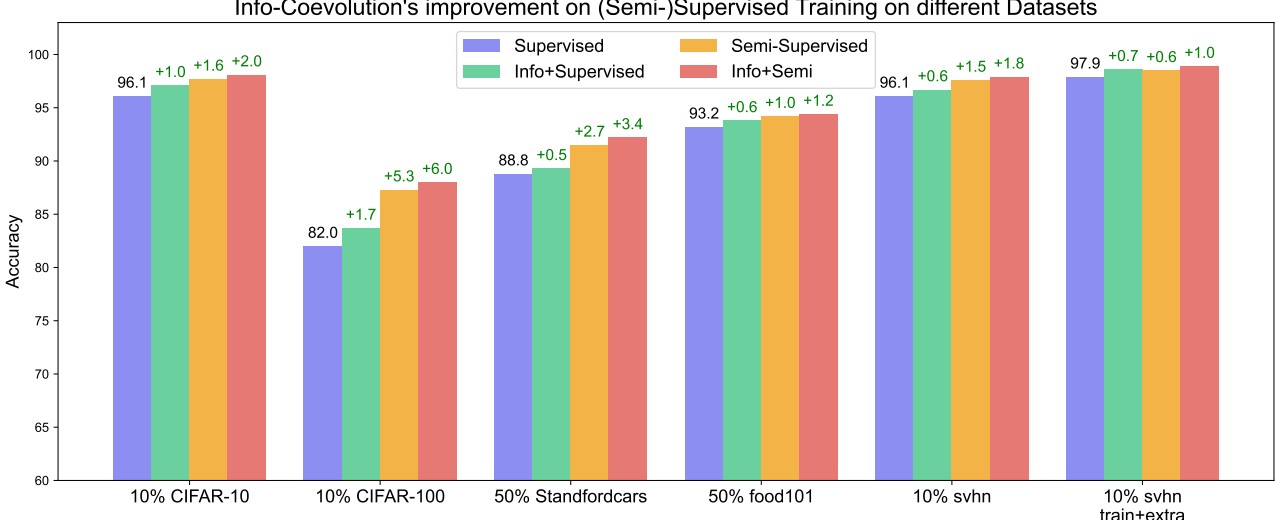

*Figure 7.* Info-Coevolution consistently improve the annotation efficiency across different datasets with both supervised/semi-supervised setting.

rechecking, we here ablate their corresponding influence on performance in Tab.4. The experiment is to choose annotation for 10% ImageNet data with 1% random data trained model. It can be seen that using only model prediction for sample gain estimation could fail to beat the random baseline (78.5% compared to 79.2%). As analyzed, purely model-uncertainty-based sample selection is unaware of distribution, which could lead to distribution bias in hard problems. Using our data's view prediction gives a better prediction of information gain and gets 0.6% performance improvement; the dynamic rechecking which introduced locality consideration effectively mitigate the problem of model-based sample selection, improving performance by +0.5% (compared to model-only -0.7%). Our proposed multiview prediction fusion combines both model and KNN prediction to better decide the samples with higher entropy. Using the fused prediction alone will get a severe distribution balance problem, which lower down the performance by 2.1%. With dynamic rechecking involved to balance both density and class balance, it gets a 1.0% ACC improvement compared to the random baseline.

**Ablation of Model Updating Frequency.** We also study the influence of model fine-tuning frequency on annotation-efficiency. In Fig.8, we can see that, when we add additional model updates during the loop, Info-Coevolution can select a better annotation set at the annotation ratio. When directly selecting 10% (0.128M) annotations for ImageNet-1k with model trained on 1% data, the performance is 80.2%; after adding two model updates in the middle, we get 80.5%.

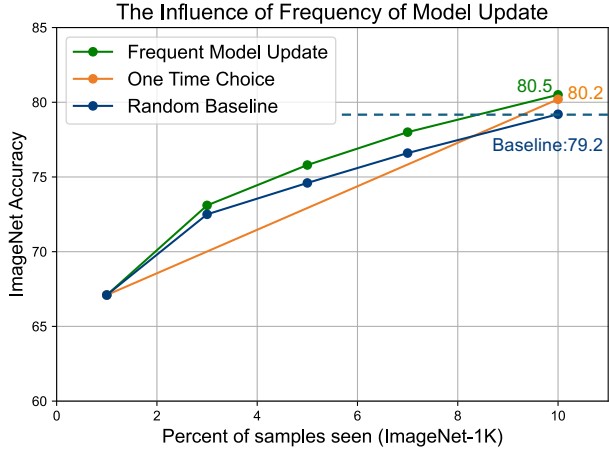

*Figure 8.* Ablation of model update frequency. When updating the model more frequently during selection, it shows improved data efficiency.

### 4.4. Task Generalization

Beyond the traditional classification task, we further extend Info-Coevolution to the semantic segmentation task. We experimented on ADE20k(Zhou et al., 2017) with UperNet(BEiT-v2 backbone)(Xiao et al., 2018; Peng et al., 2022). The UperNet has a backbone (BEiT-v2-large) and a UperHead. We use the backbone feature from the last layer (dim 1024*16*16) and mean pooling it to dim 1024. We adapt Info-Coevolution accordingly, using pixel-wise confidence average as model confidence, class-wise similarity weighted confidence as knn confidence, and $gain = 1 - avg(confidence_{model}, confidence_{knn})$. The result for using 1% data-trained model to select 10% data is shown in

Table 5. Comparison of the performance of selected 10% data on ADE20k. Our method provides a better data annotation selection.

|         | aAcc | mIoU | mAcc |
|---------|------|------|------|
| Random  | 82.19 | 46.89 | 58.72 |
| Ours    | $82.84_{\uparrow 0.65}$ | $48.39_{\uparrow 1.50}$ | $60.81_{\uparrow 2.09}$ |

table 5. Our method achieves a significantly better performance at the same data amount.

## 5. Conclusion

In this work, we extend Information Gain estimation to classification tasks with locality consideration. We proposed a novel online efficient algorithm Info-Coevolution, to increase the annotation efficiency of supervised/semi-supervised training. Info-Coevolution is able to save the annotation ratio by 32% on ImageNet with a lossless performance and is compatible with Semi-Supervised learning to achieve almost lossless performance with 50% annotation. We also demonstrate how to enhance the downstream task dataset with open-source data. As an online method, Info-Coevolution is efficient and extendable for real-world applications.

**Limitations and future works** Our work mainly considers the same training data distribution as the target distribution on the classification task. Using the large-scale weakly supervised data may be using different training data distribution to target. For unlabeled data and weakly-supervised data, it is possible to extend the framework further while we have only done a preliminary study. Tasks beyond classification are also worth further exploration.

**Acknowledgement** This work is supported by Bytedance, and was done when Ziheng Qin intern at Bytedance. Ziheng Qin, Hailun Xu (experiments) and Kai Wang (writing) are core contributors. Kanchan Sarkar and Danhui Guan are project leads. Kai Wang is supported by the National Research Foundation, Singapore under its AI Singapore Programme (AISG Award No: AISG2-PhD-2021-08-008). Yang You's research group is being sponsored by NUS startup grant (Presidential Young Professorship), Singapore MOE Tier-1 grant, ByteDance grant, ARCTIC grant, SMI grant (WBS number: A-8001104-00-00), Alibaba grant, and Google grant for TPU usage. We thank Dawei Du for valuable discussions and feedback.

## Impact Statement

This paper presents research aimed at advancing the field of efficient training in the context of continuously growing datasets. We conducted experiments using publicly available, widely-used datasets in machine learning, without incorporating any new data. As such, our work poses no risk of harmful societal consequences. Instead, it contributes to society by enabling the development of better neural models at a lower cost.

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

## A. Proof

We show our proof of Theorem3.1 here as follows: For the model $M = g \circ f$, assume $g$ is $L_g$-Lipschitz, i.e. for all $z_1, z_2$ in the feature space, we have

$$||g(z_1) - g(z_2)|| \leq L_g ||z_1 - z_2|| \tag{8}$$

Then,

$$||g(z_1) - g(z_2)|| \leq L_g ||z_1 - z_2|| \leq L_g \epsilon \tag{9}$$

Softmax is known to be 1-Lipschitz which does not further change the bound of logits.

When using cosine distance, we know that

$$cosine\_dis(z_1, z_2) = 1 - \frac{<z_1, z_2>}{||z_1||||z_2||} \tag{10}$$

If features $||z_1||, ||z_2||$ are normalized (as in ViT networks), then

$$||z_1 - z_2|| = \sqrt{2 * cosine\_dist(z_1, z_2)} \tag{11}$$

And

$$cosine\_dist(z_1, z_2) < \epsilon \Rightarrow ||g(z_1) - g(z_2)|| <= L_g \sqrt{2\epsilon} \tag{12}$$

For a single linear layer (g) with W ($\mathbb{R}^{d \times C}$) and b, its Lipschitz constant is bounded by the spectral norm of W, which is $||W||_2$. However, for a Gaussian weight with $N(0, \sigma^2)$, it could still have a bound of $O(\sigma(\sqrt{d} + \sqrt{C}))$. In practice, this is still quite big, and it only serves as a worst-case bound.

There is a tighter bound using the local gradient assuming smoothness (which is true for most type of g):

$$||g(z_1) - g(z_2)|| \leq sup_{\lambda \in [0,1]} ||\nabla g(z_\lambda)|| \cdot ||z_1 - z_2|| \leq \epsilon \cdot sup_{\lambda \in [0,1]} ||\nabla g(z_\lambda)||, \tag{13}$$

where $z_\lambda = \lambda z_1 + (1 - \lambda) z_2$.

Empirically, the linearity mainly takes effect when the predictions are reasonably good. It is because

- On one hand, when the prediction is poor, the neighbour could be more random and the predicted knn-prediction will give a high entropy itself as well as the model, which will not contradict our purpose of the algorithm (the corresponding sample would very likely have a high entropy and gain, and the theoretical guarantee of knn pred is from the k value);

- On the other hand, when the prediction is good (e.g. p¿0.7 or higher) in the region, the local gradient will be smaller, and the softmax backward propagation further suppresses the bound with $||p - y_t|| <= 1$.

So in the cases where the local gradient might fail to give a reasonable bound, the result itself would very likely have high entropy and be correctly estimated as high entropy by both model and knn, and the knn's estimation error doesn't matter too much. The actual algorithm reliance on this linearity is relaxed in bad cases due to the algorithm design. And for regions with good predictions, we can empirically evaluate the value with the local gradient.

## B. Additional Experiment Details

We follow the experimental settings in Semi-ViT. Our results are trained on a single node of 8 NVIDIA A100-SXM4-80G. For supervised finetuning, we train the model with batchsize 512, learning rate 0.001 for 50 epochs with all augmentations same as in Semi-ViT.

## C. Semi-supervised Data Selection

For selecting the semi-supervised data, we further add another term based on each sample's average distance to nearest k neighbours in high-confidence samples and already selected samples.

$$Semi - Gain(z) = \sum_{x \in KNN, x \in Selected} cosine\_dist(z, x) \tag{14}$$

This term is to encourage a more uniform distribution of unlabeled samples in the space between the labeled samples, so that the semi-supervised training can progressively learn the pseudo target.

