# OpenReview forum: "Info-Coevolution: An Efficient Framework for Data Model Coevolution"
_ICML.cc/2025/Conference — ICML 2025 poster_

### Official Review · Reviewer_i2qw · 2025-03-11

**Overall Recommendation:** 3

**Summary:**

The paper addresses the challenge of high annotation costs and inefficiency in training models on growing datasets by proposing a framework for online selective annotation that co-evolves data and models. The method combines information gain estimation (model uncertainty and dataset locality via nearest-neighbor analysis) with Bayesian prediction fusion to merge model and data-derived predictions, reducing bias, and dynamic rechecking to update sample priorities post-annotation for class balance. Using efficient approximate nearest-neighbor search (HNSW), it scales logarithmically, enabling million-scale dataset handling. Key results include achieving lossless ImageNet-1K performance with 32% fewer annotations (68% total) and 50% with semi-supervised learning, while generalizing across datasets (CIFAR, SVHN) and architectures (ViT, ResNet). The framework integrates public data (e.g., LAION-400M) to enhance performance with minimal unlabeled data and incurs low overhead (~10 GPU hours on ImageNet). It outperforms coreset selection and active learning by avoiding distribution bias and automates annotation halting when gains plateau.

**Claims And Evidence:**

Yes

**Essential References Not Discussed:**

No

**Experimental Designs Or Analyses:**

The paper only compares with one AL method, DQ (coreset-based), in Fig. 6. More AL methods could be compared, from the early entropy-based methods to recent AL methods. I understand the proposed method is both data-agnostic and model-agnostic, but it'd be better to include more AL methods to comprehensively evaluate its effectiveness.

More experiments on different semantic understanding tasks could be considered.

**Methods And Evaluation Criteria:**

To me, the proposed method is more like a hybrid active learning (AL) methods combining uncertainty-based AL (in this method, based on the decision boundary of a pretrained backbone) and diversity-based AL (in this method, based on the nearest neighbor data similarity) for information gain estimation. Therefore, I cannot read this work as a new paradigm for AL. It's more like an improvement work for hybrid AL.

Regarding the method design, information gain has been widely recognized as an effective criterion for AL. The Bayesian Prediction Fusion of data similarity and model prediction is reasonable to me. It provides to some extent new insights for AL community towards the data-agnostic and model-agnostic AL.

Experiments on ImageNet, CIFAR-10, CIFAR-100, Standfordcars, fool101 and SVHN are conducted. These are commonly used active learning datasets, so the classification-based AL experiments are generally adequate.

However, only image classification task is evaluated. Showing results on segmentation/detection tasks would be more convincing.

**Other Comments Or Suggestions:**

None

**Other Strengths And Weaknesses:**

None

**Questions For Authors:**

None

**Relation To Broader Scientific Literature:**

The proposed method would benefit ML training by reducing the annotation and training costs on large datasets.

**Theoretical Claims:**

I checked Theorem 3.1, and it looks correct.

---

> ### Author Rebuttal · Authors · 2025-03-31
>
> We sincerely thank the reviewer i2qw for the suggestions as well as the potential improvements. We make responses as follows.
>
> **Q1: The paper only compares with one AL method, DQ (coreset-based), in Fig. 6. More AL methods could be compared, from the early entropy-based methods to recent AL methods. I understand the proposed method is both data-agnostic and model-agnostic, but it'd be better to include more AL methods to comprehensively evaluate its effectiveness. More experiments on different semantic understanding tasks could be considered.**
>
>
> Thanks for the suggestions. We attach more baselines here and extend our algorithm to the semantic segmentation task.
>
> ### **More baselines**
>
> ImageNet-1k with ViT supervised training from 1% data to 10% data,
> |Method|step|Acc|Cost|
> |---|---|---|---|
> |MASE|9%|80.0|1 training + selection|
> |MASE|1w|80.2|12 training + selection|
> |BASE|9%|79.7|1 training + selection|
> |BASE|1w|80.2|12 training + selection|
> |Partial BADGE|9%|78.9|1 training + selection|
> |Partial BADGE|1w|80.4|12 training + selection|
> |Info-Coevolution|9%|**80.2%**|**1 training** + selection|
> |Info-Coevolution|~2.25%|**80.5%**|**4 training** + selection|
>
> **Analysis:**
>
> - Previous active learning methods require training for each step. At the same step number, Info-Coevolution has the best performance with better time cost. Given the full budget, other baseline methods have much higher costs ($O(n^2)$ for these step-wise methods) and still have lower performance than Info-Coevolution.
>
> - What's more, these methods are also hard to further scale (due to $O(n^2)$ training complexity) while Info-Coevolution can directly scale from 12w samples to 87w samples without a in-middle training step.
>
> - Additionaly, Info-Coevolution doesn't introduce any data distribution problems, so we are able to do continual training. We have additionally verified that using continual training on previous checkpoints has no performance loss. (1% data with 50 epoch, continual training with selected10% data with 45% epoch, then continual training on 68% data with 40 epoch) The training cost of our algorithm could be O(n) in total for selected data.
>
> **Conclusion:**
>
> - Info-Coevolution has better performance and much better scalability than previous baselines.
>
> ### **Semantic segmentation**
>
> We here extend our algorithm to semantic segmentation and attach results on ADE20k with UperNet(BEiT-v2 backbone). The UperNet has a backbone (BEiT-v2-large) and a UperHead. We use the backbone feature from the last layer (dim 1024\*16\*16) and mean pooling it to dim 1024. We adapt Info-Coevolution accordingly, using pixel-wise confidence avg as model confidence, class-wise similarity weighted confidence as knn confidence, and $gain=1-avg(confidence_{model},confidence_{knn})$. The result for using 1% data trained model to select 10% data is as follows:
>
> **ADE20K 10% random**
> |aAcc|mIoU|mAcc|
> |---|---|---|
> | 82.19 | 46.89 | 58.72 |
>
> **ADE20K 10% our selection (using BEiT feature)**
> |aAcc|mIoU|mAcc|
> |---|---|---|
> | 82.84 **$\uparrow$0.65** | 48.39 **$\uparrow$1.50** |60.81 **$\uparrow$2.09**|
>
> **Analysis:**
> - It can be seen that the **mAcc, mIoU, and mAcc are improved by 0.65, 1.50, and 2.09 respectively**, which is significant at this data amount. (For larger data ratio we are still running experiments and will update the results in the later updates. We will add thorough experiments in the revision. The code of this part will also be published.)
>
> **Conclusion:**
> - Info-Coevolution is generalizable to more tasks as analyzed.

---

### Official Review · Reviewer_9pQu · 2025-03-12

**Overall Recommendation:** 3

**Summary:**

This paper points out the current issue of active learning. Traditional active learning methods select informative data points for annotation but suffer from high computational costs, frequent model retraining, and bias in uncertainty-based selection. Considering these, they propose a novel framework called Info-Coevolution, a model-data fusion coevolution model that integrates: (1) Bayesian information gain estimation to evaluate how much information a sample contributes to model improvement; (2) kNN approximation with HNSW, to measure entropy and confidence without model retraining and (3) Bayesian fusion, to model confidence and data-driven uncertainty for more robust sample selection. The authors then test their model on CIFAR-10, CIFAR-100, ImageNet-1K, and the Info-Coevolution got 68% annotation cost while maintaining full model performance and 50% annotation cost with semi-supervised learning.

**Claims And Evidence:**

Yes.

**Essential References Not Discussed:**

See previous sections.

[r1] Ash, Jordan T., et al. "Deep batch active learning by diverse, uncertain gradient lower bounds." arXiv preprint arXiv:1906.03671 (2019).

[r2] Citovsky, Gui, et al. "Batch active learning at scale." Advances in Neural Information Processing Systems 34 (2021): 11933-11944.

**Experimental Designs Or Analyses:**

The baselines selection is too limited, just include very normal baselines like random, coreset. Can consider stronger baselines like BADGE (incorporating the uncertainty and diversity based measures, it is comparable).

**Methods And Evaluation Criteria:**

Yes.

**Other Comments Or Suggestions:**

Should use more space to describe Bayesian fusion.

Line 025-028: "For real-world datasets like ImageNet-1K, Info-Coevolution reduces annotation and training costs by 32% without performance." is not a completed sentence.

**Other Strengths And Weaknesses:**

Strengths:
1. It reduces the re-training cost;
2. It reduces annotation costs by 30-50% while maintaining performance.

Weaknesses:
1. Should add stronger baselines;
2. No ablation study to show the effectiveness of (1) Bayesian fusion, like Bayesian vs. direct calibration (e.g., temperature scaling); Bayesian fusion vs. Monte Carlo Dropout-based uncertainty estimation and remove this part; (2) kNN approximation vs. true model entropy estimation (can be replaced by computing from full retraining).

**Questions For Authors:**

See Section "Other Strengths And Weaknesses".

**Relation To Broader Scientific Literature:**

There is no discussion about: Bayesian fusion vs. standard active learning work that combines the model- and data-driven approaches like BADGE and Batch Active Learning at Scale.

**Theoretical Claims:**

For the Bayesian information gain estimation, no proof shows how kNN-based entropy estimation approximates the true situations or the model-based entropy estimates (approximation error).

---

> ### Author Rebuttal · Authors · 2025-03-31
>
> We sincerely thank the reviewer 9pQu for the suggestions as well as the potential improvements. We make responses as follows.
>
> **Q1: Adding references and baselines**
>
> **A1**:Thanks for the suggestions. We attach more baselines here, and will add the corresponding references.
>
> ImageNet-1k with ViT supervised training from 1% data to 10% data,
> |Method|step|Acc|Cost|
> |---|---|---|---|
> |MASE|9%|80.0|1 training + selection|
> |MASE|1w|80.2|12 training + selection|
> |BASE|9%|79.7|1 training + selection|
> |BASE|1w|80.2|12 training + selection|
> |Partial BADGE|9%|78.9|1 training + selection|
> |Partial BADGE|1w|80.4|12 training + selection|
> |Info-Coevolution|9%|**80.2%**|**1 training** + selection|
> |Info-Coevolution|~2.25%|**80.5%**|**4 training** + selection|
>
> **Analysis:**
>
> - Previous active learning methods require training for each step. At the same step number, Info-Coevolution has the best performance with better time cost. Given the full budget, other baseline methods have much higher costs ($O(n^2)$ for these step-wise methods) and still have lower performance than Info-Coevolution.
>
> - What's more, these methods are also hard to further scale (due to $O(n^2)$ training complexity) while Info-Coevolution can directly scale from 12w samples to 87w samples without a in-middle training step.
>
> - Additionally, Info-Coevolution doesn't introduce any data distribution problems, so we are able to do continual training. We have additionally verified that using continual training on previous checkpoints has no performance loss. (1% data with 50 epoch, continual training with selected10% data with 45% epoch, then continual training on 68% data with 40 epoch) The training cost of our algorithm could be O(n) in total for selected data.
>
> **Conclusion:**
>
> - Info-Coevolution has better performance and much better scalability than previous baselines.
>
> **Q2: Ablation study to show the effectiveness of (1) Bayesian fusion, (2) kNN approximation**
>
> **A2**: Thanks for the question. We make a clarification here, and ask for the reviewers' clarification on some question points to further clarify/discuss in the next response.
>
> We have conducted part of the ablation in sec 4.3 ablation, including kNN only, model-only, fused, and their combination of dynamic rechecking. The Data column is referring to kNN-only. It can be seen dynamic rechecking is the one guaranteed non-biased distribution when incorporating model-based gain estimation (model estimation without dynamic rechecking will lower the performance), while kNN-only without dynamic rechecking could improve performance already.
>
> For the calibration, which calibration (e.g., temperature scaling) is referred to? Could you give a reference so we can be more clear and add a comparison?
>
> Monte Carlo Dropout-based uncertainty estimation introduces random noise in the feature space, which is somehow similar to our aggregating information from nearby samples to estimate. But the differences are:
> 1. our method does not have to do inference multiple times.
> 2. As a model-based method, Monte Carlo Dropout-based uncertainty estimation is estimating the model optimization space. It could still suffer distribution problems, as the dropout is not considering data distribution (it is more about the local curvature of the model).
>
> For kNN approximation vs. true model entropy estimation, our weighted KNN-based method doesn't introduce much distribution bias. Incorporating model-based estimation usually causes this (as observed with lower performance than randomly selected data), and needs to be fixed by dynamic rechecking or other distribution trimming methods.
>
> **Q3: For the Bayesian information gain estimation, no proof shows how kNN-based entropy estimation approximates the true situations or the model-based entropy estimates (approximation error).**
>
> **A3**: We would like to first clarify on the point: is the reviewer referring to the problem that kNN-based entropy estimation or model-based entropy estimation could differ to real entropy?
>
> We add some preliminary discussions here and will update it in the next reply based on the reviewer's feedback. Empirically, the model confidence (defined in section 3) is linear correlated (more than 0.95) to sample prediction accuracy on both training data and validation data. We are using this probability proxy instead of real entropy (mentioned in sec 3), because it keeps this observed linearity instead of introducing log function with an unstable bound, and thus when linearity does hold, the knn prediction can help to decide the sample priority.
>
> We also added some discussion in A1 to Reviewer aJ9B about our Lipschitz constant assumption, which is somehow related.
>
> We are open to further discussions.

---

> > ### Comment · Reviewer_9pQu · 2025-04-05
> >
> > I mean, your kNN-based entropy estimation implicitly assumes that local neighborhoods in the feature space reflect the model’s predictive behavior, therefore, aggregating confidence from nearby samples can replace retraining-based entropy or Monte Carlo-based uncertainty estimation, right?
> >
> > So, considering it as a confidence estimator for unlabeled samples, the author should (1) compare kNN-based entropy vs. true model entropy (e.g., computed by retraining the model, and measuring entropy) and (2) compare with entropy from multiple dropout inferences (MC Dropout), Deep Ensembles, Expected Calibration Error (ECE), Trust score (To Trust Or Not To Trust A Classifier, Jiang et al. 2018).

---

> > > ### Author Response · Authors · 2025-04-08
> > >
> > > Thanks for clarifying the points. We are now more clear with the questions and make responses as follows.
> > >
> > > First, we would like to clarify that, Info-Coevolution is directly estimating the gain of labeling a sample in the confidence space without calculating entropy explicitly (it is taking advantage of the linearity discussed in section 3, while we can still approximately calculate it according to section 3.4). Using the confidence as proxy only needs to keep the most confident prediction and the corresponding confidence.
> > >
> > > Second, as stated in sec 3, Info-Coevolution is designed to conduct sample selection and predict the information gain of labeling/learning a sample, instead of being a confidence estimator for a single sample. The information gain of a sample (not the "true entropy") is estimated on a model and the data distribution, which is quite different to those confidence estimators defined on a model and a single sample, and it is not fully comparable. However, we found Bayesian Fusion actually could perform as a kind of orthogonal model calibration method to model-based ones like temperature scaling.
> > >
> > > **How kNN-based dynamic rechecking prediction approximates retrained model prediction**
> > > We evaluate the effectiveness of dynamic rechecking (with kNN-based prediction + Bayesian Fusion) on approximating model retraining with the following setting: we start from ViT model trained with our 5% ImageNet-1K data at validation accuracy 75.8%, and extend the data to 7% data.
> > >
> > > For samples updated by dynamic rechecking with an updated gain larger than 0.1, the **cosine similarity** with retrained model's gain estimation ($1-confidence$) is 0.82479. It shows a quite high correlation, as we are using a k=8 (so kNN prediction has a granularity about 0.125). A higher estimated gain also shows a higher cosine similarity (cosine similarity 0.88 for gain>0.5).
> > >
> > > This suggests that dynamic rechecking can approximate the model retraining on predicting high-gain samples efficiently and effectively.
> > >
> > > **Discussion on Model Calibration Works**
> > > As stated in the second point above, the gain estimated by Info-Coevolution's Bayesian Fusion is a different one to those of model calibrations (confidence gain on model and dataset vs. confidence of model on single sample). These model calibration methods can potentially be applied to enhance the model confidence estimation part of Info-Coevolution, introducing a corresponding cost.
> > >
> > > Table of Comparison
> > > |Method|Purpose|Cost|Range of Confidence|
> > > |---|---|---|---|
> > > |MC Dropout|Model Calibration|Inference*M|[0,1] with std|
> > > |Deep Ensemble|Model Calibration|Training*M|[0,1]|
> > > |Temperature Scaling|Model Calibration|Inference*1+One Parameter Training on Validation Set|[0,1]|
> > > |Trust Score|Model Calibration|Training Data Inference*1+KNN|[0,$\infty$)|
> > > |Info-Coevolution|Estimate Sample Annotation Gain|Training Data Inference*1+KNN|[0,1]|
> > >
> > > In terms of the cost and confidence ranges, MC Dropout and Deep Ensemble is introducing much larger cost than our origianl algorithm; Trust Score's confidence range is non-compatible. Due to these factors and time limit, we here first try wheter Temperature Scaling could helps to give a better initial confidence, and wheter Bayesian Fusion is effective with it.
> > >
> > > **Effectiveness of Bayesian-Fusion on Confidence Estimation**
> > >
> > > **Setting:** ViT model trained with our 5% ImageNet-1K data, the Expected Calibration Error (ECE) of model prediction and Bayesian fusion is (lower is better)
> > >
> > > | |Model|Bayesian-Fusion|Temperature Scaling|Bayesian-Fusion with Temperature Scaling|
> > > |---|---|---|---|---|
> > > |ECE|0.310|**0.275**|0.181|**0.171**|
> > >
> > > **Analysis:** We can see the Bayesian-Fusion better estimates the real model confidence, as it has a lower Expected Calibration Error. For the case using original model confidence and the case using temperature scaling, Bayesian-Fusion improved the expected calibration error in both cases.
> > >
> > > **Conclusion:** Though Bayesian Fusion is designed for gain estimation, it could also improve model confidence estimation as an orthogonal method with model calibration methods.
> > >
> > > **Update:** We further investigate the effect of incorporating model calibration method into Info-Coevolution framework. When selecting additional **2%** ImageNet data using ViT model trained with our 5% ImageNet-1K data, the calibrated model (by temperature scaling) could further improve our performance:
> > >
> > > |Data Amount|Ours (previous)|Ours+Temperature Scaling|
> > > |---|---|---|
> > > |5%->7%|78.0|78.9($\uparrow$0.9)|
> > > |7%->10%|80.5|80.5|
> > > |10%->50%|85.1|85.0|
> > >
> > > **Analysis:**
> > >  - The calibrated model can provide a better original confidence estimation. **When the model is not good enough** It improves the linearity gain/confidence prediction of our framework and could benefit sample selection. When the model is good, temperature scaling has a negligible benefit on improved model confidence estimation and the sample choice.

---

### Official Review · Reviewer_aJ9B · 2025-03-13

**Overall Recommendation:** 4

**Summary:**

The paper introduces Info-Coevolution, a framework for selective data collection which aims to improve data annotation efficiency. It proposes strategies to estimate information gain by leveraging Bayesian principles, and also uses ANN structures to help achieve efficient data selection with minimal computational overhead, which reduces the need for frequent model updates over the selection process. The framework was benchmarked against conventional active learning and coreset selection methods, and was shown to reduce annotation costs while maintaining model performance.

**Claims And Evidence:**

The claim that Info-Coevolution can reduce annotation and training costs without compromising model performance was also support by benchmarks on various datasets, where it was compared with other baseline methods. The ablation experiments also further validated that each component of the methodology contributes to the performance of the framework.

**Essential References Not Discussed:**

N/A

**Experimental Designs Or Analyses:**

The experimental designs and analyses were sound, they were performed on well known benchmark datasets, and compared with established baselines under supervised and semi-supervised settings, with clearly defined annotation metrics such as accuracy improvements across different annotation ratios and annotation efficiency. The ablation experiments also served to demonstrate that each component of the framework contributes to the performance improvements.

However some limitations would include that the benchmarks only contained image/computer vision datasets and the models used were mostly limited to ViTs and ResNets. The set of experiments do not demonstrate the generalizability of the method to other data modalities or models, and expanding that evaluation would make the analysis stronger.

**Methods And Evaluation Criteria:**

Yes, the methodology introduced tackles the task of improving annotation efficiency while maintaining the model performance. Benchmarks were performed on well known datasets such as ImageNet-1k, CIFAR-10/100 etc to showcase the method’s effectiveness compared to other baseline methods. The authors also evaluated both the supervised and semi-supervised learning settings, covering different training paradigms.

**Other Comments Or Suggestions:**

There is a typo in the abstract where a sentence is incomplete:

> … Info-Coevolution reduces annotation and training costs by 32% without performance.

**Other Strengths And Weaknesses:**

Strengths:
- The methodology is novel by integrating several ideas in a unified framework
- The benchmarks have demonstrated the effectiveness of the method and the savings it can achieve without compromising model performance, which is key for real world applications
- The experiments were clear and ablation study helped to break down the components of the approach

Weaknesses:
- The authors mention in the limitations section that their experiments mainly considers cases where the training and target distributions are the same, the method claims to be able to address data distribution shifts however it was not shown in the benchmarks
- Other concerns/weaknesses have been pointed out in the comments above

**Questions For Authors:**

The paper does not provide a detailed analysis of the hyperparameters used in the framework (such as the distance and similarity thresholds). Could the authors give more insight into finding the optimal values, and how adjusting them might affect the method’s performance.

**Relation To Broader Scientific Literature:**

The contributions are related to several areas in ML, notably active learning and coreset selection. Traditional active learning methods focuses on selecting the most informative samples for annotation, usually based on model uncertainty. Info-Coevolution additionally integrates distribution awareness of the data for selection, which aides in addressing common issues in active learning such as bias and distribution shifts. It also builds upon coreset selection ideas where representative subsets are chosen to improve training. But unlike traditional coreset methods which are often static, Info-Coevolution filters data dynamically and in an online manner, making the process more efficient. Lastly, the methodology extends ideas from information theory, using information gain based selection, but incorporates additional information such as data similarity and using a bayesian prediction function to estimate the information gain for each datapoint more accurately.

**Theoretical Claims:**

Yes, the proof for Theorem 3.1 is correct, however it should be noted that it is made under the assumption that the function g is Lipschitz continuous. While most common used neural networks exhibit Lipschitz continuity, the Lipschitz constant may be difficult to compute rigorously and large in practice. Hence the bound may not be tight in real world settings, a limitation that should be addressed.

---

> ### Author Rebuttal · Authors · 2025-03-31
>
> We sincerely thank the reviewer aJ9B for the recognition and appreciation of our work, and for the valuable questions as well as comments. For the comments and questions, here are our responses:
>
> **Q1: The Lipschitz constant may be difficult to compute rigorously and large in practice.**
>
> **A1**:Thanks for the good question. Previously we use the theorem 3.1 to support our introduction of locality, but it is true that the Lipschitz constant is not tight enough to be used in many real settings. We here discuss a tighter bound and the overall reasonability.
>
> For a single linear layer ($g$) with W ($\mathbb{R}^{d\times C}$) and b, its Lipschitz constant is bounded by the spectral norm of W, which is $||W||_2$. However, for a Gaussian weight with $N(0,\sigma^2)$, it could still have a bound of $O(\sigma(\sqrt d + \sqrt C))$. In practice, this is still quite big, and it only serves as a worst-case bound.
>
> There is a tighter bound using the local gradient assuming smoothness (which is true for most type of $g$):
> $$\lVert g(z_1)-g(z_2)\rVert_2 \leq sup_{\lambda \in [0,1]} \lVert\nabla g(z_\lambda)\rVert_2 \leq sup_{\lambda \in [0,1]} \lVert \nabla g(z_\lambda)\rVert_2 \cdot\epsilon ,$$ Where $z_\lambda = (1-\lambda)z_1 + \lambda z_2$ .
>
> Empirically, the linearity mainly takes effect when the predictions are reasonably good. It is because
> - On one hand, when the prediction is poor, the neighbour could be more random and the predicted knn-prediction will give a high entropy itself as well as the model, which will not contradict our purpose of the algorithm (the corresponding sample would very likely have a high entropy and gain, and the theoretical guarantee of knn pred is from the k value);
> - On the other hand, when the prediction is good (e.g. p>0.7 or higher) in the region, the local gradient will be smaller, and the softmax backward propagation further suppresses the bound with $||p-y_t||<=1$.
>
> So in the cases where the local gradient might fail to give a reasonable bound, the result itself would very likely have high entropy and be correctly estimated as high entropy by both model and knn, and the knn's estimation error doesn't matter too much. The actual algorithm reliance on this linearity is relaxed in bad cases due to the algorithm design. And for regions with good predictions we can empirically evaluate the value.
>
> **Q2: The method claims to be able to address data distribution shifts however it was not shown in the benchmarks**
>
> **A2**: The data distribution shifts in the article refer to the problem of model-based active learning: it only emphasizes on samples learned poorly by the model, and the selection causes a distribution problem and worse performance. This is shown in the ablation as dynamic-rechecking/KNN-only are all without this performance drop. We will revise the corresponding parts to make this clear.
>
> **Q3: The set of experiments does not demonstrate the generalizability of the method to other data modalities or models, and expanding that evaluation would make the analysis stronger.**
>
> **A3**: Thanks for the advice. We further add semantic segmentation experiment on ADE20k with UperNet(BEiT-V2 backbone). See A2 to Reviewer X845.
> For modalities other than vision, if there is a sample-level feature, it is also possible to extend the framework to them. Will add more discussion in the revision of future work.
>
> **Q4: Typo in the abstract**
>
> **A4**: Thanks, we have fixed it in local revision. Should be "… Info-Coevolution reduces annotation and training costs by 32% without performance loss".
>
> **Q5: Hyperparameters used in the framework (such as the distance and similarity thresholds). Could the authors give more insight into finding the optimal values, and how adjusting them might affect the method’s performance.**
>
> **A5**: The hyperparameters should depend on the feature space itself (how dense is the data in the space), and can use empirical methods to evaluate(retrieve some samples to estimate the knn distance distribution and knn-prediction correlation, and an adequate k).
>
> Generally, k = 8, cosine similarity 0.9 is good enough, and cosine similarity 0.85 also works in the setting; 0.95 will be too big as there will be very few near neighbour pairs.
>
> Too small distance (too large similarity threshold) may fail to reduce cost and become a model-based method (for dynamic rechecking. Actually using 0.9 for knn and 0.85 for dynamic rechecking is also a good choice), while too large distance threshold may affect sparse-region samples' knn-prediction (their k-near-nighbour could be further and less linear correlated, and should be kept for the sake of generalization).
>
> In our actual use cases, the threshold value is not sensitive, cosine similarity 0.9 directly works without tuning, and cosine similarity 0.85 doesn't make a statistically significant difference.

---

### Official Review · Reviewer_X845 · 2025-03-14

**Overall Recommendation:** 2

**Summary:**

The paper presents **Info-Coevolution**, a framework aimed at enhancing the co-evolution of data and models through **online selective annotation**. The primary goal is to minimize annotation costs while preserving model performance by utilizing **Bayesian Prediction Fusion** and **data locality analysis** to assess the information gain of samples. This approach selectively annotates data, facilitating efficient dataset construction and model training.

Key findings include:
- **Reduced Annotation Costs**: Info-Coevolution achieves **lossless performance** on ImageNet-1K with only **68% of the annotation cost**, further reducing to **50%** with semi-supervised learning.
- **Efficiency**: The framework incurs minimal computational overhead, completing the selection process in **1 minute** for million-scale datasets.
- **Compatibility**: It is compatible with both supervised and semi-supervised learning, avoiding distribution shifts during continual training.

The paper also investigates **retrieval-based dataset enhancement** using unlabeled open-source data, showing improved performance with additional unlabeled data.

**Claims And Evidence:**

The claims are generally well-supported:
- **Reduced Annotation Costs**: Supported by experiments on ImageNet-1K, CIFAR-10/100, and other datasets, demonstrating comparable or superior performance with fewer annotations.
- **Efficiency**: Backed by computational overhead analysis, indicating logarithmic scaling and rapid completion for large datasets.
- **Compatibility with Semi-Supervised Learning**: Validated through experiments with Semi-ViT and Fixmatch, showing enhanced data efficiency.

However, the claim of **generalizability to other tasks** is not fully substantiated, as experiments are confined to classification tasks.

**Essential References Not Discussed:**

The paper does not discuss **dataset distillation**, a highly relevant field that shares similarities with the goals of Info-Coevolution. Dataset distillation focuses on synthesizing a small, informative dataset that can be used to train models with performance comparable to training on the full dataset. This is conceptually aligned with Info-Coevolution's goal of reducing annotation costs while maintaining model performance.

A key work in this area is:
- **Wang et al. (2018)**: "Dataset Distillation" (arXiv:1811.10959). This paper introduces the concept of dataset distillation, where a small synthetic dataset is created to mimic the performance of a larger dataset. The methods and insights from this work could provide valuable context for Info-Coevolution, particularly in terms of reducing data requirements while preserving model performance.

The omission of this reference is notable, as dataset distillation represents a complementary approach to the problem of efficient dataset construction and could enrich the discussion of related work in the paper. Including this reference would help situate Info-Coevolution within the broader landscape of data-efficient machine learning techniques.

**Experimental Designs Or Analyses:**

The experimental designs are robust and validate key claims:
- **ImageNet-1K Experiments**: Demonstrate lossless performance with 68% annotation cost, compatible with semi-supervised learning.
- **Comparison with Coreset Selection**: Shows competitive performance with state-of-the-art methods.
- **Generalization Across Datasets**: Consistent improvements in annotation efficiency on CIFAR-10/100, StanfordCars, and other datasets.

However, experiments are limited to **image classification tasks**, lacking validation on **real-world industrial data** or **larger datasets**. The computational overhead, while low, remains significant for resource-limited settings.

**Methods And Evaluation Criteria:**

The methods and evaluation criteria are suitable for the problem:
- **Bayesian Prediction Fusion** and **data locality analysis** are well-justified for estimating information gain and enhancing sample selection efficiency.
- The use of **Approximate Nearest Neighbor (ANN)** structures like HNSW ensures scalability for large datasets.
- Evaluation criteria (e.g., accuracy on ImageNet-1K, CIFAR-10/100) are standard benchmarks, facilitating comparison with prior work.

The evaluation could be improved by including **additional tasks** and **real-world datasets** to demonstrate broader applicability.

**Other Comments Or Suggestions:**

- **Experiments**: Expand to encompass other tasks, such as object detection and segmentation, and incorporate real-world datasets.
- **Limitations and Future Work**: Include a broader discussion on limitations and future directions, specifically regarding applicability to weakly supervised and non-classification tasks.
- **Dataset Distillation**: Discuss its relationship with Info-Coevolution to strengthen contextual grounding and demonstrate awareness of related approaches.

**Other Strengths And Weaknesses:**

**Strengths:**

- **Originality**: The integration of Bayesian Prediction Fusion, data locality analysis, and online selective annotation presents a novel and creative approach.
- **Significance**: This framework addresses the critical challenge of reducing annotation costs in machine learning while maintaining performance.
- **Clarity**: The paper is well-written with clear explanations of the methodology and results.

**Weaknesses:**

- **Limited Task Scope**: The experiments focus solely on image classification, and the method's potential application to other tasks, such as object detection and segmentation, is not investigated.
- **Theoretical Depth**: The theoretical derivations require further expansion, particularly regarding limitations and assumptions.
- **Omission of Dataset Distillation**: There is no discussion of dataset distillation, which is closely related to the goals of Info-Coevolution, aiming to synthesize a small, informative dataset that achieves performance close to training on the full dataset. This aligns with the goal of reducing annotation costs while maintaining model performance in Info-Coevolution.

**Questions For Authors:**

1. **Generalizability**: Can Info-Coevolution be extended to tasks beyond image classification, such as object detection or segmentation?

2. **Theoretical Limitations**: While the theoretical claims are plausible, they seem somewhat elementary. Could the authors provide a more comprehensive theoretical analysis concerning deep neural networks, including aspects of optimization and generalization theory?

3. **Real-World Validation**: Has Info-Coevolution been evaluated on real-world industrial datasets or in online annotation environments?

4. **Dataset Distillation**: How does Info-Coevolution compare to dataset distillation methods, such as those proposed by Wang et al. (2018) and subsequent works?

[1] Wang et al. (2018). "Dataset Distillation" (arXiv:1811.10959).

**Relation To Broader Scientific Literature:**

The paper builds on and extends prior work in:
- **Active Learning**: Addresses limitations of traditional methods by integrating model-specific estimation with distribution awareness.
- **Coreset Selection**: Improves upon existing methods by leveraging model-specific information without requiring fully annotated data.
- **Semi-Supervised Learning**: Bridges the gap between fully supervised and weakly supervised approaches.

The paper contributes to the literature by proposing a **more efficient and scalable approach** to data annotation and model training, with potential applications in resource-constrained settings.

**Theoretical Claims:**

The paper presents several theoretical claims:
- **Theorem 3.1**: Asserts that predictions for nearby samples in feature space are similar under certain distance thresholds. The proof, provided in the appendix, appears correct but relies on assumptions (e.g., Lipschitz continuity) that warrant further discussion.
- **Information Gain Estimation**: The extension to broader tasks is theoretically sound, but the derivation could benefit from more rigorous analysis, especially in multi-class settings.

In general, while the theoretical claims are credible, they also appear somewhat trivial.

---

> ### Author Rebuttal · Authors · 2025-03-31
>
> We sincerely thank the reviewer X845 for pointing out the missing references as well as the potential improvements. We make responses as follows.
>
> **Q1: Dataset Distillation references not discussed.**
>
> **A1**: Thanks for the advice. In general, dataset distillation and active learning have a main overlapping research area which is coreset selection (if not requiring full annotation in advance, then the coreset selection can also serve as active learning), and is introduced in related works. Dataset distillation methods also include synthetic data methods, which is another setting not comparable in this work. We will add a discussion of dataset distillation to the related works to make it more clear.
>
> **Q2: Task Scope beyond image classification such as object detection and segmentation, is not investigated.**
>
> **A2**: Thanks. We here extend our algorithm to semantic segmentation and attach results on ADE20k with UperNet(BEiT-v2 backbone). The UperNet has a backbone (BEiT-v2-large) and a UperHead. We use the backbone feature from the last layer (dim 1024\*16\*16) and mean pooling it to dim 1024. We adapt Info-Coevolution accordingly, using pixel-wise confidence avg as model confidence, class-wise similarity weighted confidence as knn confidence, and $gain=1-avg(confidence_{model},confidence_{knn})$. The result for using 1% data trained model to select 10% data is as follows:
>
> **ADE20K 10% random**
> |aAcc|mIoU|mAcc|
> |---|---|---|
> | 82.19 | 46.89 | 58.72 |
>
> **ADE20K 10% our selection (using BEiT feature)**
> |aAcc|mIoU|mAcc|
> |---|---|---|
> | 82.84 **$\uparrow$0.65** | 48.39 **$\uparrow$1.50** |60.81 **$\uparrow$2.09**|
>
> **Analysis:**
> - It can be seen that the **mAcc, mIoU, and mAcc are improved by 0.65, 1.50, and 2.09 respectively**, which is significant at this data amount. (For larger data ratio we are still running experiments and will update the results in the later updates. We will add thorough experiments in the revision. The code of this part will also be published.)
>
> **Conclusion:**
> - Info-Coevolution is generalizable to more tasks such as segmentation as analyzed.
>
> **Q3: Theoretical Depth: requires further expansion, particularly regarding limitations and assumptions.**
>
> **A3**: Thanks for the advice. The current theoretical part supports us to introduce locality into the algorithm, which could greatly improve the efficiency of distribution-based entropy estimation methods, and benefit efficient data re-balance (dynamic data rechecking).
>
> We agree an expansion should be in the assumption of Lipschitz-continuous, please refer to the discussion in the reply A1 to reviewer aJ9B Q1 (due to char limit this year). We are open to further discussion.
>
> On the other hand, the current assumption is that the target distribution should be between IID and uniform, so any sample predicted as redundant by the framework doesn't affect the final classification result. We will attach a theoretical discussion in the next update for this argument. So a limitation is that it currently doesn't handle a target distribution beyond the range of IID to uniform. A solution could be adjusting the sampling frequency based on target distribution.
>
> All the discussed parts will be added to the revision, and we are **open to further discussion** for the theoretical part.
>
> **Q4: Include a broader discussion on Limitations and Future Work (applicability to weakly supervised and non-classification tasks)**
>
> **A4**: Thanks. The semantic segmentation experiment is now added in A2. We will add a more detailed discussion in the revision.
>
> In general, the framework could be applied to models with an $f$, which produces a feature for each sample (and could be used for a retrieval task). If mutual information could be approximated as in classification, it could be a good usage case; even if not a direct approximation, the ANN could also provide distribution-based estimation (with a weighted mean of uncertainty).
>
> Classification can be interpreted as a kind of coarse-grained retrieval.
>
> For weakly supervised training, as for CLIP/BLIP schema, they are retrieval tasks directly training the $f$ with contrastive training. The locality is also taking effect, and mutual information could be estimated in the feature space. So theoretically our framework is also applicable to weakly supervised training, and the reannotation gain could be estimated for human/ChatGPT annotation to improve data quality.
>
> **Q5: Real-World Validation**
>
> **A5**: We evaluate the method with some in-house data pipeline, and it achieves promising results.

---

### Decision · Program_Chairs · 2025-05-01

**Decision:**

Accept (poster)

**Comment:**

This paper introduces Info-Coevolution, a framework for efficient dataset construction and training through online selective annotation. Info-Coevolution enables coevolution of the model and dataset using task-specific models. The method selectively annotates streaming or web data, significantly reducing annotation and training costs without degrading performance.

The paper presents an important contribution to efficient learning at scale. While one reviewer raised concerns about the theoretical underpinnings, these were satisfactorily addressed in the rebuttal and discussion. The approach is novel, the results seem strong and relevant, and the broader community would benefit. I recommend acceptance and encourage the authors to incorporate reviewer feedback in the final version.